# Neural Load Disaggregation: Meta-Analysis, Federated Learning and Beyond

Hafsa Bousbiat [1,†] , Yassine Himeur [2,†] , Iraklis Varlamis [3,†] , Faycal Bensaali [4,*] and Abbes Amira [5]

1   Department of Informatics, University of Klagenfut, 9020 Klagentfurt, Austria
2   College of Engineering and Information Technology, University of Dubai,
    Dubai P.O. Box 14143, United Arab Emirates
3   Department of Informatics and Telematics, Harokopion University of Athens, Tavros, 177 78 Athens, Greece
4   Department of Electrical Engineering, Qatar University, Doha P.O. Box 2713, Qatar
5   College of Computing and Informatics, Sharjah University, Sharjah P.O. Box 27272, United Arab Emirates
*   Correspondence: f.bensaali@qu.edu.qa
†   These authors contributed equally to this work.

**Abstract:** Non-intrusive load monitoring (NILM) techniques are central techniques to achieve the energy sustainability goals through the identification of operating appliances in the residential and industrial sectors, potentially leading to increased rates of energy savings. NILM received significant attention in the last decade, reflected by the number of contributions and systematic reviews published yearly. In this regard, the current paper provides a meta-analysis summarising existing NILM reviews to identify widely acknowledged findings concerning NILM scholarship in general and neural NILM algorithms in particular. In addition, this paper emphasizes federated neural NILM, receiving increasing attention due to its ability to preserve end-users' privacy. Typically, by combining several locally trained models, federated learning has excellent potential to train NILM models locally without communicating sensitive data with cloud servers. Thus, the second part of the current paper provides a summary of recent federated NILM frameworks with a focus on the main contributions of each framework and the achieved performance. Furthermore, we identify the non-availability of proper toolkits enabling easy experimentation with federated neural NILM as a primary barrier in the field. Thus, we extend existing toolkits with a federated component, made publicly available and conduct experiments on the REFIT energy dataset considering four different scenarios.

**Keywords:** load disaggregation; neural NILM; federated learning; energy recommender systems

## 1. Introduction

Global energy demand is rising quickly, which in turn, makes the need for electric energy rise even faster, especially in household setups. Current studies reveal that the most crucial element in resolving energy issues would be the intelligent and cost-effective use of electricity as the primary source of energy [1]. This, in turn, raises the need for systems that recommend best practices and actions to use energy in homes, workplaces, and buildings more efficiently [2–4]. To recommend positive actions to the users and help them adopt a more efficient energy consumption behavior, it is essential first to capture their energy footprint and analyze their behavior concerning the use of appliances [5,6]. The analysis of energy utilization can help in this regard. A viable solution is to use smart meters and sensors to record the energy consumption of each appliance, potentially combined with smart data analytics to visualize the energy consumption habits [7]. Nonetheless, a more financially affordable solution is to use only a single meter, and non-intrusive load monitoring (NILM) techniques [8,9] to identify the consumption of each appliance from the aggregate measurements. NILM techniques offer thus the possibility to determine which appliances are utilized in a household at any moment and the corresponding amount of

energy consumed [10]. Therefore, these approaches can be leveraged by different services such as activity monitoring [11], and the detection of defective appliances [12].

Several algorithms were suggested to address the NILM problem [13]. Nonetheless, deep neural networks have received significant attention since their first introduction [14]. These models fast became the main research stream in NILM scholarship, mainly encouraged by the availability of several real and synthetic energy datasets (e.g., REFIT [15], SynD [16]), enabling the training and testing of these models considering different scenarios. Many of these approaches were developed to evaluate the advantages and drawbacks of different deep learning concepts on the energy disaggregation task and achieved significant enhancement in the performance.

Despite these recent advancements, several open research questions remain unaddressed. For example, the transferability of these models remains problematic for real deployments in smart grids. Furthermore, the computational complexity of neural NILM approaches requires high processing power. Consequently, the majority of these models are implemented on the cloud. Leveraging pruning and compression techniques to reduce the models' size is a viable solution in this regard [17] where NILM models can operate on the edge. On the one hand, using local models would prevent updating the model with new data measurements from other clients. On the other hand, uploading the consumers' data to the cloud would lead to privacy and security concerns from the consumer side, which was identified as a primary obstacle to the acceptance of smart metering technology [18]. To overcome these issues, federated learning (FL) has recently been leveraged to train and share NILM models [19,20] providing a solution to both of the previous concerns.

The vast and fast-increasing attention that the NILM research topics received in the last decade created the need for more systematic reviews summarising recent advancements in the field. Consequently, not only the number of NILM contributions has increased in recent years, but also the number of systematic reviews published yearly. This trend is yet to rise, considering the current energy crisis and the potential of NILM approaches in mitigating its effect on the end consumer.

This work contributes to the current literature by performing an in-depth analysis of existing NILM reviews and highlighting their contributions. The main advantages and drawbacks of neural NILM models are identified by discussing the different learning paradigms that can mitigate the computational complexity or redistribute it more efficiently between the cloud and edge nodes. Furthermore, we evaluate the federated NILM models' performance in a simulated energy measurements dataset. Furthermore, the current paper suggests an FL disaggregator fully compatible with the most recent NILM toolkits (i.e., NILMtk [21] and Deep-NILMtk [21]). We publish the code of the disaggregator to enable and facilitate further studies and research on the topic. To summarize, the main contributions of this article are as follows:

- presenting an umbrella review that discusses existing NILM systematic reviews and identifies their contributions;
- analyzing recent FL-based NILM studies and highlighting their advantages and drawbacks;
- identifying open FL-based NILM challenges and deriving directions where actual and near future research, works are heading;
- introducing an FL-based disaggregator fully compatible with NILMtk.

The rest of this paper is organized as follows: In Section 2, we explain the methodology adopted to conduct the proposed umbrella review of NILM systems. In Section 3, we present an overview of the different findings revealed in recent NILM reviews. In Section 4, we provide an overview of the main data engineering approaches covered in recent NILM systematic reviews. Next, Section 5 describes relevant NILM algorithms, comparison, and evaluation setups. Section 6 discusses the FL paradigm with an overview of the leading frameworks available in the scope of NILM. Following, a case study simulating a distributed training environment is presented in the case of the REFIT dataset considering different scenarios in Section 7. Thereafter, Section 8 identifies FL-based NILM open

challenges and derives future research directions. Finally, Section 9 concludes the current paper with a summary of our study's main contributions and limitations.

## 2. Methodology

There has been an increasing interest in NILM in the last decade leading to an increasing number of published contributions. Consequently, many systematic reviews have already been published. Figure 1 illustrates the number of published NILM reviews per year, considering only the Scopus database. Accounted alone, 2022 witnessed the publication of ten (10) NILM reviews. With this increased number of systematic reviews available, a logical next step is to review existing systematic reviews, allowing the findings of separate reviews to be compared and contrasted, thereby establishing a broader overview of the topic.

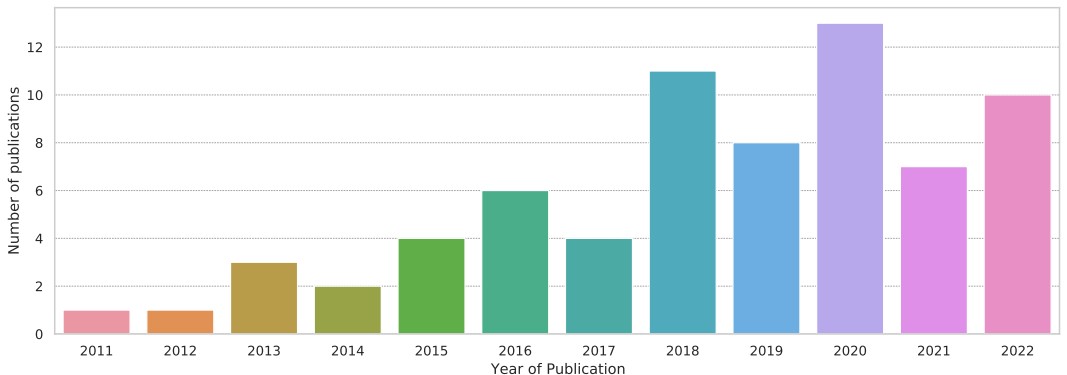

**Figure 1.** The number of published NILM reviews per year.

For this purpose, an umbrella review was adopted in the first part of the current study following the protocol suggested in [22]. Four databases were queried, Scopus, Google, Scholar, IEEE explore, and ACM library with the key terms Survey, review, NILM, and load disaggregation. Our search was further limited to reviews that were published in English during the last two years to include the most recent findings about the topic. A total number of twenty-two reviews was obtained. A first screening step based on the title and the abstract was established to consider only reviews with the main focus on NILM scholarship and eliminate reviews about related topics or only subtopics (e.g., only datasets [23]). In a second step, the methodological quality of the reviews included was assessed using an adapted version of the AMSTAR checklist [24], which resulted in considering only ten reviews published during the last two years. The main goal of the meta-analysis study is to address the following research questions:

- **RQ1.1**: What taxonomies exist in the literature for NILM approaches?
- **RQ1.2**: What are the main NILM-related topics that have been covered extensively (or merely) in recent reviews?
- **RQ1.3**: What are recent reviews' common findings?
- **RQ1.4**: What open research gaps exist in the literature, and what potential future directions have been identified so far?

Among others, the topic of FL was identified as one of the topics that have merely been covered in existing reviews. The reviews presented in [9,25] were the only ones addressing this topic. Yet, it was only superficially discussed. In addressing this gap in the literature, a systematic literature review of available federated NILM frameworks was established in the second part. The same four databases considered in the first part were queried. Nonetheless, only fourteen papers addressing the topic, all from the past two years, were obtained, which further highlights the novelty of the topic. The systematic review was conducted with the following research questions in mind:

- **RQ2.1**: What aspects of FL have been investigated in the case of NILM?

- **RQ2.2**: What are the state-of-the-art performances of federated NILM approaches?
- **RQ2.3**: What are the major challenges facing the adoption of this learning paradigm in NILM scholarship?

Furthermore, we identify the non-availability of NILM toolkits suitable for federated training of disaggregation models as a major obstacle in evaluating FL in NILM scholarship. Thus, we extend available toolkits with a federated trainer fully compatible with the API of NILMtk and present a simulation study on a single appliance from a real energy dataset to address the following research questions:

- **RQ3.1**: What is the effect of the data amount on the disaggregation performance?
- **RQ3.2**: What is the effect of the number of local iterations on the disaggregation performance?

### 3. Findings from the Meta-Analysis

Table 1 summarises the main topics covered in the ten NILM reviews selected during the current study. It is clear from the table that some topics were extensively reviewed while others gained less attention. The learning algorithms and existing data sets with metrics have been among the most investigated topics. Data engineering topics (i.e., feature extraction, preprocessing, and postprocessing), existing toolkits, and computing platforms gained less attention. Interestingly, only a handful of reviews performed a comparative study of existing approaches with only two qualitative and four quantitative comparisons. Most of these studies leveraged values reported in the reviewed contributions to provide recommendations for future directions. While leveraging values reported in different contributions to perform a comparative study does not provide rigorous findings since they result from different evaluation setups, it remains one of the most straightforward means of providing recommendations for future work. Yet, the findings of these comparisons are to be interpreted with caution. We highlight that the previous observations are highly related to the selected set of systematic reviews, which is the main limitation of the current study that the authors admit.

In summary, the literature and research landscape of NILM is full of algorithms, solutions, and systems, which can be broadly categorized into the following domains, as depicted in more detail in Figure 2:

- **Data**: Data are the primary concern and, at the same time, the main asset of NILM approaches, particularly recent ML/deep learning (DL) approaches. All the necessary data-related tasks in all other ML and data mining problems appear in the case of energy load data: preprocessing, feature extraction and management, and representation are the most important, and the decisions made in each one of them respectively affect the NILM quality.
- **Algorithms**: Despite the majority of reviews briefly discussing traditional NILM algorithms, there is a consensus to put the main focus on ML algorithms. More precisely, the disaggregation algorithms have evolved during the last three decades from simple combinatorial optimization to ML models and, more recently, to DNN models, which are now considered state-of-the-art algorithms.
- **Computing platforms**: The choice of the platform where the NILM computations will take place is equally essential to the choice of algorithms or data. The choice can be application dependent, but it also must consider more constraints related, for example, to the requested efficiency and the need for privacy. Cloud computing is the primary choice when computational performance is the priority. However, edge and fog architectures are gradually gaining ground to cover the need for privacy and distributed processing.

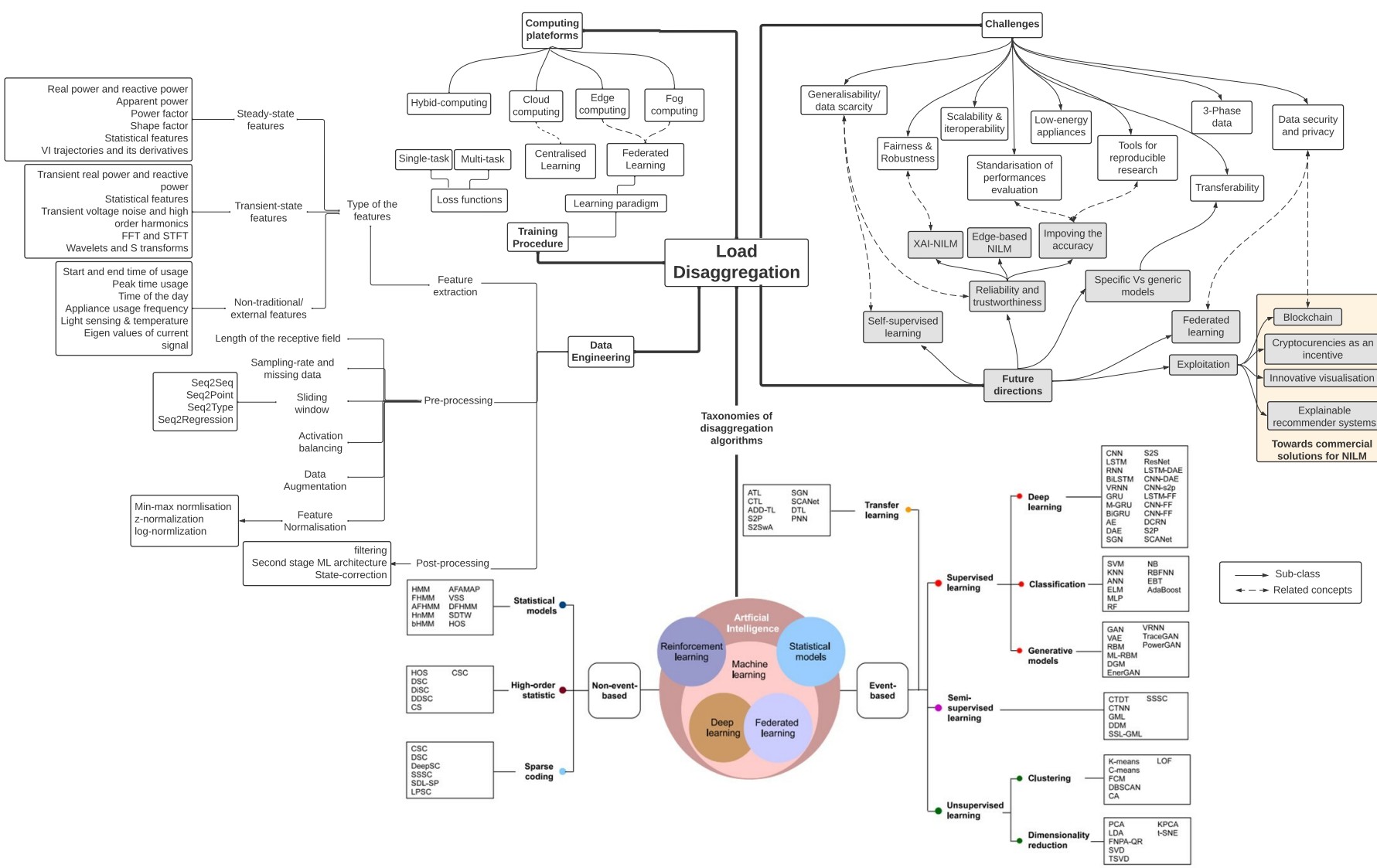

**Figure 2.** Overview of NILM concepts included in the reviewed reviews (extended from [9]).

- **Training process**: Since the algorithms have evolved into trainable models, the process that we follow to train them has become a critical part of NILM. The choice of process is strongly connected to the underlying computation platform. Since the distributed paradigm with edge nodes that carry computational tasks is now feasible, FL paradigms are gaining hype over their centralized alternatives.
- **Challenges**: The choice of one or the other platform, algorithm, learning paradigm, and data strategy may solve several issues but can also set important challenges that must be handled to optimize the NILM quality, efficiency and performance.
- **Future directions**: A complete study of the NILM landscape must also comprise an analysis of the next steps. These future directions can be based on recent advances and future directions in related research domains such as AI and ML.

In the following sections, we discuss the above-mentioned concepts, providing more details on the algorithms proposed in the last decade for neural NILM while summarising the findings from different reviews. We also focus on the learning paradigm of federated NILM, emphasizing its advantages and the challenges that are still open. To the best of the authors' knowledge, the current manuscript is the first to provide an overview of federated NILM frameworks. At the end of this article, we discuss the future directions of Neural and Federated NILM, taking into account the overall trends in AI and ML research and projecting them onto the NILM industry.

**Table 1.** Main concepts included in recent NILM reviews.

|  | [26] | [9] | [27] | [28] | [25] | [29] | [30] | [31] | [32] | [33] | Our Study |
|---|---|---|---|---|---|---|---|---|---|---|---|
| **Basic NILM concepts** | | | | | | | | | | | |
| Definitions | ✓ | | | | ✓ | | ✓ | ✓ | | ✓ | |
| Problem formulation | ✓ | | ✓ | | | | ✓ | ✓ | | | |
| **Algorithms design** | | | | | | | | | | | |
| Feature extraction | | ✓ | ✓ | ✓ | ✓ | | | ✓ | ✓ | | ✓ |
| Preprocessing | ✓ | | ✓ | | | | | | | ✓ | ✓ |
| Postprocessing | | | | | | | | ✓ | | ✓ | ✓ |
| Learning algorithm | ✓ | ✓ | ✓ | ✓ | ✓ | ✓ | ✓ | ✓ | ✓ | ✓ | ✓ |
| FL | | ✓ | | | ✓ | | | | | | ✓ |
| **Evaluation** | | | | | | | | | | | |
| Datasets | ✓ | ✓ | ✓ | | ✓ | ✓ | ✓ | ✓ | | ✓ | ✓ |
| Metrics | ✓ | | ✓ | | ✓ | ✓ | ✓ | ✓ | | ✓ | ✓ |
| Toolkits | ✓ | | | | | ✓ | | | | | ✓ |
| Computing platforms | | ✓ | | | | | | ✓ | | | ✓ |
| **Comparative study** | | | | | | | | | | | ✓ |
| Qualitative | ✓ | | | | | | | ✓ | | | |
| Quantitative | | | | | ✓ | ✓ | | ✓ | | ✓ | |

## 4. Data Engineering for NILM

The emergence of ML algorithms in NILM scholarship highlighted the importance of data engineering to enhance the disaggregation performance for different appliances. Thus, this aspect received particular attention from recent systematic reviews. The current manuscript groups three main data processing steps under the term data engineering: data preprocessing, feature extraction, and postprocessing techniques. Nonetheless, feature extraction received relatively more attention from the selected set of reviews than preprocessing and postprocessing techniques adopted in different contributions.

The preprocessing techniques adopted in different contributions were covered in two reviews, mainly [27,33]. The authors of [27] highlight two main techniques considered mandatory for the majority of algorithms: (i) handling sampling rates and missing data, and (ii) balancing. The first technique, handling sampling rates and missing data, is related to the quality of the data sets during training and is leveraged to address potential technical problems that may occur in real setups (i.e., hardware and communication issues). The second technique provides a balance between the ON states/events of each appliance

and the OFF states/events. The latter problem is mainly caused by residential appliances being OFF most of the time. In addition to the previous two techniques, the authors of [33] provided an overview of data augmentation techniques adopted mainly to address the underrepresented classes.

An overview of the types of features in NILM was suggested in five different reviews, mainly [9,25,28,31]. A consensus between all these reviews can be concluded where three types of features were highlighted: steady-state features, transient features, and external/nontraditional features. We emphasize that all three types of features are handcrafted features. Further details about each type are provided in Figure 2. Alternatively, the reviews presented in [27,32] provide a classification of NILM features based on the sampling frequency required, where they offered a clear distinction between low-frequency and high-frequency features, as follows:

- High-frequency sampling: This approach involves collecting data at a high rate, such as at a rate of one to several times per second [32]. This can provide a high level of detail and resolution, leading thus to improved accuracy. The majority of transient features require high sampling rates.
- Low-frequency sampling: This approach involves collecting data at a lower rate, such as at a rate of once per minute or once per hour. This can be less resource-intensive but may also result in a lower level of detail and accuracy.

Considering post-processing techniques, only reviews presented in [31,33] provided an overview of existing approaches for NILM algorithms. One of the main findings of the quantitative analysis provided by the first review (i.e., [33]) was the enhancement that can be achieved, where they found that 28% to 54% of improvement was recorded in related work. Consequently, it is to conclude that postprocessing techniques are a key factor in improving existing algorithms. In this regard, Figure 2 illustrates three techniques that were suggested for signal postprocessing [31] in the case of NILM. Yet, further research is required on existing postprocessing techniques for NILM and their potential taxonomies.

It was widely acknowledged in all the reviews that ML and AI models are the most prominent algorithms in the NILM scholarship in recent years. Consequently, data engineering techniques are of enormous importance to future NILM developments. Nonetheless, our analysis reveals that recent reviews only focused on the feature extraction phase while providing less attention to preprocessing and postprocessing techniques, even though there is evidence that these two steps can enhance the final performance.

## 5. NILM Algorithms, Comparison, and Evaluation Setups

The non-intrusive monitoring of the operation and energy consumption of appliances, especially in household setups that consist of a large variate of loads and specific usage patterns, has been recognized as an essential task for more than three decades, with the seminal work of Hart that defined the task [8]. Since then, several techniques have been proposed, broadly categorized into the major groups depicted in Figure 3.

The first group of solutions was mostly based on *combinatorial optimization* techniques, which assumed that the total load was the result of a combination of appliances (with known loads) that operate in different states (even not operating at all) and tried to find the combination of appliances and states that better matches the overall load measurement. Taking this one step further, *hidden Markov models* **(HMM)** attempted to model the task using a probabilistic approach concerning the appliances that operate at every moment and the state they are on. In the last decade, technological advancements in neural networks and the underlying infrastructures that support their operations, as well as the abundance of training data, gave rise to the ML approaches for NILM and mainly to neural NILM, which demonstrated state-of-the-art performance under a variety of training conditions (e.g., high sampling rates, enough computational capacity).

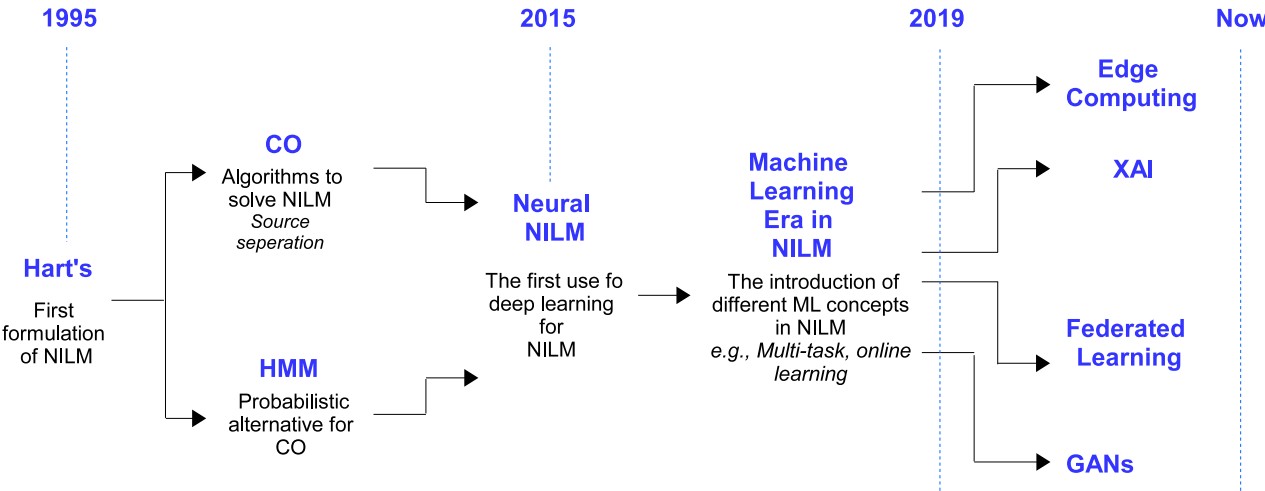

**Figure 3.** Historical development of NILM.

The problem of non-intrusive monitoring of appliances' load based on the disaggregation of the measurements from a single monitoring device is usually approached in the literature by breaking it into smaller tasks. Given a known inventory of appliances for a household, these tasks comprise (a) the detection of different states for each appliance, (b) the extraction of signatures per state and appliance, and (c) the classification of each measurement to the most promising combination of appliances' states [34]. Instead of monitoring the operation of each appliance on a second-by-second basis, some NILM techniques simply identify state change events and consequently record the start and end time of an appliance usage and the total energy consumed [35]. Alternatively, Neural NILM models provide a point-to-point solution for each appliance.

Convolutional neural networks (CNNs) can be employed to detect state-change events. As suggested in [35], a current sequence of length $L^2$ is transformed in an image of $L \times L$ pixels and is fed to a CNN, which is then trained to identify appliances initially on a single load task. This task allows distinguishing between appliances when a single appliance is on at each moment. This is taken one step further by establishing a multi-load identification task, in which the model is trained to distinguish between all possible load combinations. The main restriction of such approaches is that the number of appliances in a household can be large. Consequently, the respective number of combinations that must be identified at any moment becomes huge.

Energy measurement data are usually considered to be in the form of time series or sequences. Consequently, the respective DNN architectures that capture the temporal semantics of input have also been employed. More specifically, recurrent neural networks (RNNs) have been used in [36] as an alternative to combinatorial optimization. RNNs successfully reconstruct the appliance signatures for the aggregated measurements and can perfectly fit appliances they have already been trained on. However, they need help to generalize on unseen appliances or power states and require vast amounts of data and a lot of computational power to be trained. In an attempt to improve the generalization of RNNs, authors in [37] employ gated recurrent units (GRU) and show that they outperform the RNN baseline. In the same direction, authors in [38] suggest using LSTM-RNNs to tackle the vanishing gradient problem better whilst learning the long-term patterns that constitute the appliances' signatures in the multi-state and multi-appliance setup.

The autoencoders (AEs) represent another architecture commonly used to extract more coherent input data representations. As such, they can be used to extract the features that compose the signature of the various appliances. They are composed of encoding and decoding layers, and at training time, they learn to optimize the output so that it better resembles (if not identical) the input. After training, the encoder is used to obtain the representation of the input to a different dimension. A stochastic variation of autoencoders is the

denoising autoencoders (dAEs), which introduce noise to the input so that the autoencoder does not learn the identity function (i.e., f(x) = f) during training. Consequently, the energy disaggregation task can be approached as a denoising problem, utilizing techniques that can transfer a noisy overall consumption from multiple appliances to a "clean" consumption of each individual appliance, using as input either active, reactive, apparent power, current, voltage, or any combination of them.

Denoising AEs employs a 1-D convolutional layer in the encoder part to feed the input measurements in segments (few seconds windows) and another 1-D convolutional layer in the decoder, with the size that depends on the size of the appliance activations [39]. They can be trained using synthetic datasets that combine the measurements of various appliances and aim to reconstruct each appliance's signature in the output. Authors in [40] have combined dAEs with RNNs to combine the merits of ANNs and HMM-based methods. Using dAEs, they obtain the signatures of the appliances, and by feeding them to the LSTM, they can identify the most promising combination of appliances (and modes) that corresponds to the aggregated consumption at any moment.

The review presented in [33] on the DNN approaches for low-frequency NILM begins with the increased requirements for processing high-frequency NILM data and continues with the evaluation of various NN-based techniques that combine CNNs with LSTMs, GRUs, and other RNN variations or even with generative adversarial networks (GANs) and AEs (denoising or variational autoencoders) in an attempt to improve the classification accuracy of collective appliance signals. The main challenge for the different algorithms relates to the overall performance, which is usually affected by the dataset used, the sampling frequencies, the input features, the metrics used for evaluation, etc. The choice of the best parameters for all the above can significantly affect the final performance as much as the architecture. According to [21], a best practice for developing DNN models is the automation of hyper-parameters tuning and selecting the appropriate architecture. Using toolkits that aggregate multiple alternative architectures allows for finding the best solution at each NILM setup.

Table 2 provides an overview of the main dilemmas, as titled in [27], or degrees of freedom, as titled in [33], existing for NILM algorithms. Further considerations were also revealed in the considered reviews. Nonetheless, we only present the ones that appeared in a maximum number of reviews to be able to contrast their findings to highlight what is widely acknowledged. Three main design choices were found to be controversial in the NILM scholarship. We provide the advantages and drawbacks of each one of them in Table 2. Furthermore, we summarise the findings reported in different reviews whenever available. Other considerations included supervised vs. unsupervised [27], RNN vs. CNN [27,33], causal vs. non-causal [27,33], sequence-to-point vs. sequence-to-Sequence [27]. These conclusions and findings remain limited since they were achieved through a direct comparison of reported values in different contributions and should be considered with a high amount of caution.

The evaluation of NILM algorithms is generally performed using widely acknowledged ML metrics and NILM datasets. Nonetheless, some evaluation metrics dedicated only to NILM models can also be identified [41] though receiving little attention in recent NILM reviews since they are less commonly used. However, despite their seldom use, these metrics could show a better summary of disaggregation results since they focus on the NILM problem by design. NILM datasets also received significant attention from existing reviews where the sampling rate and the data quality remain the main concern. Furthermore, NILM toolkits are an important part of the evaluation as they improve research efficiency. This aspect was only covered in two reviews [26,31] revealing that available NILM toolkits emphasize the algorithms without considering the available hardware and network infrastructure, which is critical for the real-time monitoring of appliances. In this direction, lightweight models [42] that combine CNNs for learning features and simple classifiers to detect appliances seem to be promising solutions. Another solution for scalability is using FL approaches [43], which can move the processing load from a centralized

to a decentralized approach taking advantage of several low processing power devices to solve the same task. Federated NILM solutions can also support privacy since data are not shared across nodes or with a centralized server [44], but also open new challenges for researchers, which are discussed in more detail in the following section.

**Table 2.** Main NILM dilemmas as discussed in recent reviews [9,26–28,31].

| | Classification | Regression | Multi-Target | Single Target | Event-Based | Eventless |
|---|---|---|---|---|---|---|
| Advantages | Reveals On/OFF states | Do not require thresholding | Reduce the training time Computationally efficient | Custom model per appliance | Efficient with high sampling rates Training can be avoided when needed | Suitable for all types of appliances Good generalization with DL model |
| Drawbacks | Require thresholding approaches | Data complexity | Convergence issues due to different operational characteristics | Computationally expensive, no minimization of the sum between real aggregate and predicted targets | Require complex hardware, scalability issues | Most algorithms require a huge amount of data for training |
| Findings unveiled in NILM reviews | Limited conclusions due to the high dependency of the performance of the classification approach on the preprocessing step. | | Simultaneous learning provides more robust models, majority of best models are multi-target, multi-target drastically reduces the computational burden | | Eventless are better for commercialization, Eventless DL demonstrated the best generalization abilities, Event-based are relatively superior considering performance but highly dependent on the event detector. | |

## 6. Federated NILM

FL [45–47], also referred to as collaborative learning, is a learning paradigm that Google introduced in 2017 to protect the privacy of its clients. Following this learning paradigm, the model is sent to the client rather than the data uploaded to a cloud server. Figure 4 illustrates the main steps of the learning process. It starts in a central server responsible for initializing the model's weight and sharing them with the clients. Upon the reception of the global model, each client executes a training task using its local data for a number of iterations and sends the new weights of the model back to the central server. Once the central server has received the local models, it will aggregate them to obtain an updated version of the global model. The process is repeated for several rounds until convergence is achieved. The most popular aggregation algorithm is known as the FedAvG [48], which relies on calculating the average of the weights of local models as an aggregation mechanism. The weighted average can be used when the size of local datasets differs for clients participating in the training. Several variants of this scheme exist in the literature, considering different aspects [46]. For example, peer-to-peer FL enables direct clients' communication and eliminates the central node [45]. More precisely, each client broadcasts their model to the other clients contributing to the training round. Considering this variant of FL, the goal is to achieve a fully decentralized training process without the need for a central server considered a single point of failure. Other variants of FL also exist but remain out of the scope of the current manuscript.

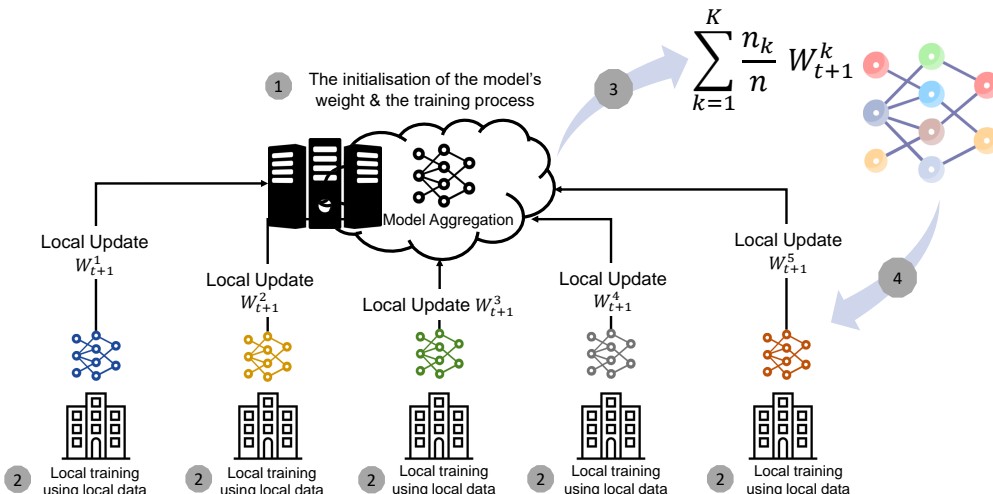

**Figure 4.** FL process.

The upgrade of the electrical grid in many countries around the globe, with the advanced metering infrastructure and edge devices, offers the possibility of adopting an FL paradigm for efficient grid management. It was extensively adopted in the case of load forecasting (e.g., [49]) and power generation prediction for renewable energies (e.g., [50]). Nonetheless, only a handful of contributions have explored the adoption of this learning paradigm in NILM scholarship: ten contributions for residential load disaggregation, one for solar energy disaggregation, and only one for investigating security aspects of FL in smart grids with respect to load disaggregation. Table 3 summarises the ten studies that investigated FL in the case of residential load disaggregation.

**Table 3.** FL frameworks for residential load disaggregation

|  | Main Contribution | Model | Dataset | Tested Appliances | Metric | Best Results | | Worst Results | | Limitations |
|---|---|---|---|---|---|---|---|---|---|---|
| [51] | Distributed federated load disaggregation with domain transferability | Seq2Point | REFIT REDD UKDALE | KTL MW DW WM FRZ | MAE | KTL | 7.23 | FRZ | 24.25 | Reported only the MAE |
| [52] | Address the problem of co-modeling in NILM | GDBT | REFIT REDD UKDALE | WM DW FRZ MW | MAE | MW | 6.89 | FRZ | 18.86 | Costly communication overhead |
| [53] | A novel aggregation algorithm for nested tasks learning | GRU RNN | Pecan Street | DRY OVEN AC EV | F1 | EV | 0.83 | Oven | 0.14 | Convergence issues |
| [54] | Extensive comparative evaluation of FL- NILM | Seq2Point | UKDALE | FRZ DW WM | F1 | FRZ | 0.78 | WM | 0.41 | Tested only on data from training buildings |
| [44] | Model's compression and collaborative learning with personalisation | SeqPoint | REDD REFIT | WM FRZ DW MW | F1 | FRZ | 0.58 | WM | 0.32 | Very low F1-scores |
| [55] | FL-based NILM focusing on both the utility optimization and the privacy-preserving by incorporating differential privacy | TEMPORAL POOLING | UKDALE REFIT REDD | Frz DW WM | F1 | WM | 0.43 | DW | 0.13 | Tested only on data from training buildings |
| [56] | Optimal model selection prior aggregation to enhance performance | Seq2point | REFIT | WM KTL DW DRY MW | F1 | MW | 0.96 | DRY | 0.87 | Tested only on data from training buildings |
| [57] | Evaluation of the noise effect of differential privacy on the performance | Seq2point | UKDALE REDD | FRY MW KTL | F1 | FRZ | 0.86 | KTL | 0.22 | Tested only on data from training buildings |
| [20] | Evaluation of a decentalized FL framework | Seq2Point | REFIT | DW MW FRZ KTL WM | MAE | MW | 0.025 | DW | 0.07 | No evaluation of the gain on the communication bandwidth |
| [19] | A new short deep NILM model for FL | Short Seq2Point | UKDALE | DW MW FRZ WM | MAE F1 | DW DW | 20 0.58 | MW MW | 61 0.10 | A limited number of nodes used for training |



An FL framework for NILM was suggested in [51], where transfer learning was used between different domains. The goal of the contribution was to protect consumers' privacy and overcome the problem of non-identically distributed data. Three public data sets were considered during the evaluation setup, where the main focus was to establish a comparison with centralized load disaggregation schemes. The results showed high potential for the suggested FL approach. Nonetheless, transfer learning from one domain to another one demonstrated poor results and showed that fine-tuning is required. Despite the extensive evaluation of the disaggregation performance, the previous study provided no analysis of the communication cost and model efficiency, and little attention was given to the hardware requirements of the edge devices. These limitations were also admitted in [56] and highlighted as future direction. Furthermore, the authors stressed the need to upgrade NILM toolkits with federated/decentralized trainers, enabling further research in this respect. Both of the previous studies adopted a Seq2Point model, which shows the strength of this model in the case of FL for load disaggregation. More precisely, even short versions of this model provide very competitive results as demonstrated in [19] where the authors suggested shortening the Seq2Point baseline trained following an FL paradigm revealing promising results despite the low number of training clients. A similar study focusing on transfer learning was suggested in [53], where a model-agnostic meta-learning approach was introduced to enable task-specific learning and allow data owners to adjust the models based on the tasks. In this regard, the FL is augmented with a meta-learning step at each round. The evaluation setup demonstrated enhanced disaggregation performance but with a longer time required for convergence.

The FL was further tested in combination with differential privacy in [57] where the Seq2Point [14] baseline was leveraged during the experimental setup. The evaluation showed that this combination provides good results in the case of the fridge, which exhibits a period consumption pattern but failed in the case of hand-operated appliances, mainly the kettle and microwave, which are directly related to daily routines. Furthermore, they demonstrate that differential privacy causes poor results due to the noise added where smaller epsilon values allow mitigating privacy attacks. Still, higher values provide similar privacy leakage to the standard FL framework. A similar study was presented in [55], evaluating the impact of noise added on the overall disaggregation of a standard federated NILM framework. The evaluation was performed using a temporal pooling model on three different data sets. It resulted in the amount of added noise drastically hindering the disaggregation task, thus achieving similar conclusions to work presented in [57].

The performance of a classification federated NILM algorithm was investigated in [54], combining FL with state-of-the-art NILM models for state classification. An extensive evaluation was conducted during this study and demonstrated competitive results. However, it mainly concentrated on using testing data from the same buildings included in the training, which may have led to biased conclusions. A multi-target federated NILM was suggested in [44]. The proposed framework leverages a multi-target learning paradigm to train a single model for all the target appliances with pruning techniques to compress the model. The experiments on three real datasets demonstrated an acceptable trade-off between privacy and disaggregation performance but with a relatively low performance, mainly a low f1-score.

Interestingly, a federated decision tree algorithm was designed in [52] for load disaggregation leveraging a two-state voting process and node-level parallelism for co-modeling NILM. During the model training phase, the server receives the local training results. It makes the final decision to select the model parameters used to split the tree nodes, including features and the corresponding thresholds. The local clients are responsible for data preprocessing, tree structure initialization, gradient computation, local histogram establishment, local split finding, and model updating. The voting thus results in a list of top-K candidate features chosen based on the maximum variance gain on local machines forwarded to the central server that will select candidate features based on majority voting.

Unfortunately, designed this way, the algorithm suffers from privacy leakage of partial feature indexes.

Despite the interesting findings of previous studies, a shared shortcoming between is their high on to the central node. More precisely, all previously presented works adopt a client-server architecture where the server represents a single point of failure. To overcome this issue, a fully decentralized FL approach was evaluated in [20] by adopting a circle topology instead of a star topology to optimize clients' communication. The experimental setup highlighted equivalent results to the centralized FL approach. However, the author did not evaluate the gain/loss in the communication bandwidth in the case of the decentralized FL. Furthermore, each node in the circle topology is a point of failure. Further research is thus required to develop a mechanism that allows to re-establish the circle in the case of failures.

The best and worst results reported for each framework are illustrated in Table 3. The results obtained on unseen buildings were chosen whenever available. Moreover, the F1-score is the most common metric among the different contributions. It is clear from the table that the results drastically differ between appliances. The highest f1-score was reported in the case of the washing machine upon optimal model selection before the aggregation in [56]. Meanwhile, the worst value was reported for the case of the dishwasher in [55]. Apart from indicating the low quality of FL frameworks, these results highlight the tremendous challenge that training on several appliances from different buildings could impose. The low values reported in [44,55] are linked to the approaches added to the standard federated framework, that is, compression and differential privacy. Overall, the reported results are acceptable, especially in the case of approaches that consider training data from different buildings and were tested on unseen buildings, simulating thus the most realistic scenario.

To the best of the authors' knowledge, all NILM toolkits are only based on centralized training. Therefore, it becomes challenging to experiment with existing implementations of different models using a federated framework since it requires extra-coding efforts. To address this gap, we extend NILMtk with a disaggregator that allows the simulation of federated training setups. The suggested extension is fully compatible with the new API of the toolkit [58]. It allows specifying the different buildings contributing to the training as well as the number of randomly selected clients in each training round. Figure 5 illustrates the different steps followed by the suggested code. Furthermore, the FedAvg algorithm is used for aggregating the locally trained models. Thus, the suggested code assumes that the same data are available for different clients. The full version of the code can be found in [59]. The disaggregator is compatible with all seq2point models implemented in [21], where the model type can be specified as a hyperparameter for the FL framework.

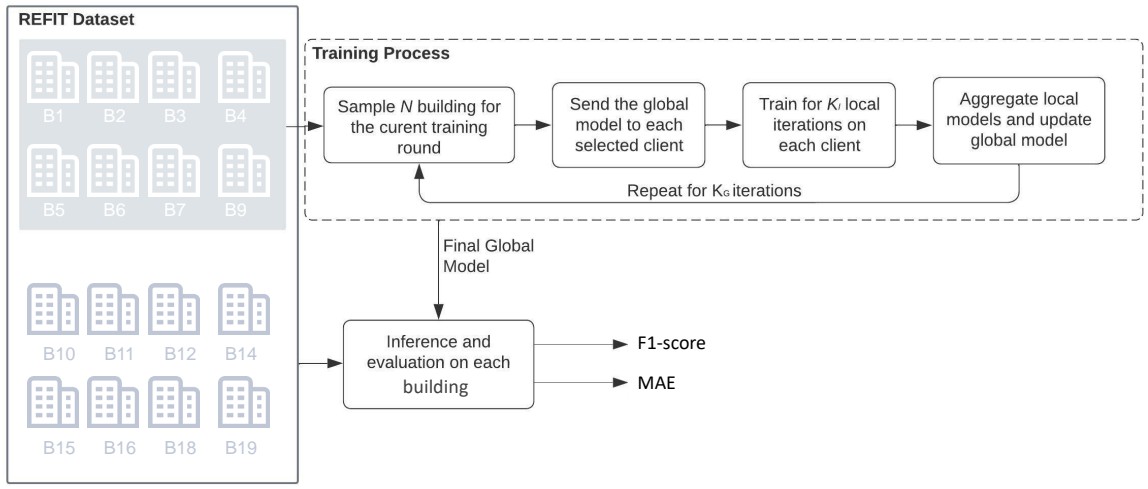

**Figure 5.** Flowchart of the simulated scenarios.

**7. Case Study**

*7.1. Data Preprocessing and Evaluation Method*

The study considers data from the REFIT [15] dataset, which contains data from 20 households in the UK. For each household, the data set contains both sub-metered data as well as aggregated data sampled every 8 s. We consider 8 households as clients during training to simulate a small energy community. Furthermore, a subset of the data measured over 15 days is used for the local training of each client. The tests are performed considering a single appliance, the fridge. The choice of this appliance is justified by its wide availability in the buildings of the considered dataset, which aligns with the goal of our study and the designed scenarios.

Before training, the data was preprocessed following several steps. First, the data are normalized using a z-normalization. Afterward, a sliding window is used with a window length of 13 min, following a sequence-to-point learning paradigm. Finally, the data are split into training and validation data for each client (85% for training and 15% for testing). Furthermore, the global model is tested on the eight households contributing to the training and eight households not included in the training subset, with one household used for validation during training.

Two evaluation metrics are used to assess the performance of the global model at the end of the simulation; mean average error (MAE) and the *F*1-score. The metrics are defined as follows:

$$\text{MAE} = \frac{1}{N} \cdot \sum_{t=0}^{N-1} |\hat{y}_t - y_t| \tag{1}$$

where, $y_t$ is the actual power consumption, $\hat{y}_t$ is the estimated power consumption, and $N$ represents the number of samples. As an absolute measure, MAE is reported in *Watts*.

$$F1 - score = \frac{2 \cdot Precision \cdot Recall}{Precision + Recall} \tag{2}$$

where the *Precision* = TP/(TP+FP), *Recall* = TP/(TP + FN). Moreover, the thresholds used to define the confusion matrix for the considered appliance were fixed to 50 *watt*.

*7.2. Simulation Setup*

The simulation study is conducted on a computer with a 3.3 GHz AMD Ryzen processor and 16 GB of memory with NVIDIA GeForce RTX 3060. Moreover, the implementation leverages the federated NILM disaggregator previously described. Despite the importance of hyper-parameter optimization, this aspect was not a primary concern in the current study. We are rather interested in evaluating the potential of adopting FL in NILM scholarship, considering large-scale data. Instead, a widely acknowledged model was adopted, that is, the Seq2Point [14] model that demonstrated outstanding performance compared to state-of-the-art models [58]. Furthermore, the standard FedAVG [60] algorithm is considered for the aggregation of local models, where stochastic gradient descent (SGD) is used as an optimizer with a learning rate of 0.0001 and momentum of 0.9. Based on previous findings in related work [58], the Seq2Point converges after 50 epochs. Thus, the number of training global rounds has been set to 10, with 5 and 10 local training epochs. All the code and the experimental setup are made available in a public repository [59].

*7.3. Experimental Results*

7.3.1. Evaluated Scenarios

The overall FL framework is described in Figure 5, along with the different steps included in the training process. Furthermore, we adopted four different scenarios summarised in Table 4. In each round, only a subset of all the clients is selected for training. The selection process is performed based on a random selection. The table illustrates the number of clients selected in the training subset of each round for the four federated scenarios. Furthermore, we will evaluate the effect of local training epochs on performance.

In summary, the goal is to assess the effect of two main aspects on the training: (i) the effect of more extensive data and (ii) the effect of the number of local training iterations. To the best of the authors' knowledge, this paper is the first to investigate the effect of these two aspects on disaggregation performance.

**Table 4.** Evaluated scenarios.

| Scenario | Clients in Subset | Local Epochs |
|:---:|:---:|:---:|
| 1 | 4 | 5 |
| 2 | 8 | 5 |
| 3 | 4 | 10 |
| 4 | 8 | 10 |

### 7.3.2. Results of the Global Model

The different scenarios evaluated during this study resulted in a set of global models that were obtained following an FL process. The evaluation results of these models in terms of MAE and F1-score are illustrated in Table 5.

**Table 5.** Disaggregation results for different buildings of the REFIT dataset.

| | Buildings of the Training Subset | | | | | | Buildings Not in the Training Subset | | | | | |
|:---:|:---:|:---:|:---:|:---:|:---:|:---:|:---:|:---:|:---:|:---:|:---:|:---:|
| | MAE | | | F1-Score | | | MAE | | | F1-Score | | |
| | Min | Mean | Max | Min | Mean | Max | Min | Mean | Max | Min | Mean | Max |
| 1 | 20.9 | 32.5 | 47.1 | 0.41 | 0.54 | 0.69 | 11.7 | 30.5 | 43.2 | 0.25 | 0.49 | 0.64 |
| 2 | 29.4 | 37.5 | 49.3 | 0.37 | 0.55 | 0.73 | 17.7 | 34.1 | 44.7 | 0.38 | 0.56 | 0.69 |
| 3 | 23.7 | 34.7 | 51.3 | 0.41 | 0.54 | 0.69 | 12.9 | 31.4 | 42.8 | 0.29 | 0.54 | 0.67 |
| 4 | 25.4 | 34.3 | 47.7 | 0.50 | 0.57 | 0.74 | 27.8 | 30.0 | 43.5 | 0.38 | 0.54 | 0.66 |

The first part of Table 5 summarises the results obtained for different scenarios in the case of clients participating in the training rounds. As the table illustrates, the values obtained in different scenarios are acceptable considering the low number of communication rounds adopted in the experimental setup compared to related work and the fact that the data are not Independent and identically distributed (non-IID), i.e., the data are not drawn from the same distribution. Notably, the global models are able to fit certain clients more than others, which can be clearly perceived when contrasting the minimum and maximum values for each scenario. When comparing scenario 1 and scenario 3, which differ in terms of the number of clients included in the training set, a slight enhancement in the F1-score is clearly observed with a higher number of clients. This observation, however, is not valid in the case of the MAE. Similar conclusions can be established when comparing scenarios 3 and 4. One reason behind this observation could be the heterogeneity of the fridge's signature between different buildings of the REFIT dataset in terms of duration and magnitude, which influences the learning process. One possible solution to overcome this obstacle could be the adoption of min-max normalization to scale all the values in the range of 0–1 before training instead of a z-normalization. This trend is further confirmed in the case of new clients (i.e., clients not included in the training set), where the models of all scenarios demonstrated good generalisability performance. These findings are interesting since they demonstrate that using only a small subset of the data would provide comparable results to more clients, leading to a more efficient resource allocation on edge.

Despite the fact that a higher number of clients with a higher number of local iterations provide better results, the case of unseen buildings from Table 5 clearly shows that adopting a higher number of local iterations could be leveraged in cases where there are constraints on the communication infrastructure to achieve equivalent results with a smaller number

of clients. The previous conclusion is more evident in the case of clients not included in the training subset as measured per scenarios 2 and 3.

## 8. Challenges and Future Directions

Despite all these advancements in NILM scholaship, many challenges are yet to address. We provide in Figure 2 the main challenges depicted from the considered NILM reviews, along with potential future directions where existing relations in a dashed line. In addition to these challenges, we identify several other research gaps in NILM research that are detailed in the following.

One of the main challenges of NILM methods is to respond in a time comparable to intrusive methods that employ sensors and smart plugs. This need for ***real-time disaggregation*** of energy consumption information requires the handling of large amounts of data in an efficient manner. The main issue for NILM systems is to develop efficient algorithms that tackle the difficulties occurring in each step of the NILM process, from data collection to estimate the power consumed by each appliance at every moment. The individual challenges in each step of the process need thorough examination and are detailed in the following.

The NILM process begins with ***data collection***, which strongly affects the quality of the final results. The higher the collection frequency, the better the disaggregation quality. This is mainly because, in frequencies below 1 Hz, only steady-state features can be extracted, which makes it difficult to distinguish between appliances with very similar consumption profiles (i.e., states with similar power values). Measuring in higher frequencies (above 1 Hz) allows the extraction of transient features, which can help to identify the small differences between such appliances, mainly occurring in a transient state or when the appliance switches among different states. The combination of steady-state and transient features compose the power signature of each appliance and can be very helpful in NILM tasks [61]. Nonetheless, the previous sampling rates are not common in real power grids where the data are generally collected every 15 or 30 min. This aspect needs thus to be investigated more by either adapting existing models or developing new models for real deployments.

Increasing the sampling frequency can improve the disaggregation quality but introduces computational and communication burdens to the NILM process, which has to be considered. Data congestion will become a problem for communication networks whenever the data frequency exceeds 1.7 Hz. Consequently, the advantages of NILM will be significantly diminished if the communication bandwidth is increased due to the ***high-cost of communication networks***. The data communication issues constitute an important challenge, especially for cloud-based NILM methods, which need large amounts of data to be transmitted and thus require high bandwidth connections. A solution to this problem can be processing aggregate energy usage data at the edge [44]. The latter solution can also be used to overcome the ***security and privacy issues*** that relate to the collection and transfer of energy usage data to the cloud. Although the aggregated data reduce the exposure of private information related to specific appliance usage, there is still a place for privacy breaches [62]. Examples include the ability to infer home occupancy, to expose consumers' appliance usage habits [63], or even their interests when very high sampling rates are employed [64].

The second step of the NILM process refers to the ***detection of appliance events***, from turn-on and turn-off to mode and continuous state changes. The main challenges in this step relate to the correct detection of the event time of occurrence and the correct distinction between appliances, especially in composite situations. Appliances, having a high fluctuation to the steady-state power during their on-state, are hard to identify. Similarly, devices with a long transition state between modes, or at least longer than the disaggregation method expects, are also hard to detect. Both ***high fluctuation*** and ***long transition*** are operation features of electric or electronic appliances. They can either be learned using lots of training data or captured using several heuristics. They constitute the

signature of an appliance, but given the wide range of brands, models, etc., it is tough to detect them properly during NILM.

If we consider a household with ***multiple and diverse appliances*** that can be turned on or off (or change state) at any moment, even ***simultaneously*** (e.g., using a multi-plug) then the task of NILM becomes even harder. If, in addition, we assume that the same NILM model is used to detect events in multiple households and the fact that each household has its composition of appliances (or appliance combinations), then more efficient and adaptive disaggregation algorithms are needed. Such algorithms must quickly adapt to the household setup and provide real-time event detection after a short load monitoring and analysis period. Furthermore, appliance transfer learning (ATL) and domain transfer learning (DTL) are required to exploit these models in smart grids. However, the transferability of these models remains challenging and problematic. This aspect was carefully investigated in [9] listing recent contributions with ATL or DTL concepts. Alternatively, unsupervised approaches can be used as suggested in [27]. A detailed discussion of transfer learning in the case of NILM is presented in [31]. As highlighted in the latter, transferability approaches are a recent research stream in NILM scholarship. Thus further research is required to address this aspect and improve the testing accuracy on unseen buildings.

To distinguish between various types of appliances, NILM systems employ special features extracted from the power consumption footprints of individual appliances. Such features, which compose the appliance's signature for each state change action, capture how the appliances work or change their states and are connected to the internal components of each appliance (such as motors and resistors). These steady-state features can be connected with the different states of the appliance, such as the active or reactive power, the current, and the voltage when the appliance is on, off, on standby, or in any other state. Although they can be easily measured, the challenge is to detect the different states of the device before creating its signature. Clustering and pairing methods are used to extract the appliance signatures and find the patterns of appliances used at every moment [65]. Another group of features related to the transition between two states, the transient features, are also gaining attention in NILM tasks. Still, they require higher sampling rates and more computational power to process and extract.

The more modern and proficient NILM methods, instead of directly applying heuristics to extract features and detect events from the measurement data, have evolved towards learning patterns (i.e., features) that can then be employed to learn the significance of each appliance. This is closer to the notion of representation learning [66,67] and can take advantage of DL architectures, such as encoder-decoders, to split the event detection step into two sub-steps: the feature learning and the detection of state changes.

The high computational demands of DL techniques instantly turned researchers to high-performance computing architectures and to cloud-based approaches, where learning and inference tasks can be carried out more efficiently. However, the deployment of cloud-based NILM algorithms for large loads of measurement data imposes more challenges for companies in the domain. First of all, large data loads require ***increased computational resources*** to be processed, which in turn blows up the cost of computational energy and the total carbon emissions [68]. Cloud-based methods cannot be completely real-time since we need time to transfer and process data to the cloud. Usually, the NILM systems' working flow works by transferring hourly power measurement segments from the metering device to a cloud database and from there to the algorithm, which further reduces the time efficiency of the approach.

NILM methods, either heuristic-based or DL based, either supervised or unsupervised, require processing lots of data and computational resources for training and inference. The open challenge in the direction of state change event detection using neural NILM is efficiently using the available resources to solve the task in a resource and cost-effectively. Edge-based NILM methods that use compressed or light models [69], trim the feature space [70], and employ hardware-based implementations on FPGAs. Other MCUs [71,72] seem to be a viable solution toward distributed NILM. The main restriction of edge-based

NILM implementations is that their models are only used for inference and not for learning, which means that either the models are re-trained on the cloud or trained only once and never adapt to changes in the household appliance load. In the former case, data must be sent to the cloud for training, thus increasing the risk of privacy exposures. In the latter case, when a new appliance is added to the household inventory or when devices are moved or combined, the model may fail to adapt to the changes.

It is unarguable that adopting FL could address some of the previously presented gaps. Nonetheless, its application in the NILM industry still is in its infancy, and many aspects are yet to address, including (i) disaggregation performance, (ii) the nonavailability of appliance-level data, and (iii) security aspects. In regard to the disaggregation performance, the fact that local energy datasets of different clients are not identically distributed (i.e., not drawn from the same data distribution [46]) remains a major obstacle to achieving state-of-the-art performance compared to centrally trained models. The reason behind this issue is consumers' heterogeneous routines reflecting directly on their energy consumption [11]. A careful look at the literature of FL [45–47] would reveal several techniques that can help mitigate this effect, including robust aggregation algorithms and clustering of the clients. While the first was only investigated in [53] with a limited experimental setup, the second technique has not been covered in the existing literature. Future work could investigate the adoption of hierarchical aggregation schemes leveraging the clustering of clients with similar load profiles to enhance the disaggregation performance. Furthermore, scholars have given little attention to the effect of local training iterations on disaggregation and the computational resources of edge devices.

Considering the nonavailability of sub-metered data in households, a handful subset of clients could be chosen on a voluntary basis to contribute to the training. Despite the fact that the previous solution is viable, it could induce high costs and maintenance requirements. Furthermore, ground truth data are commonly only available for specific appliances (e.g., fridge, washing machine) attached to a smart plug. However, other appliances, such as HVAC systems, are more energy-consuming and would be more interesting to identify. In this case, a more straightforward solution would consist of adopting a semi-supervised learning paradigm leveraging publicly available energy datasets to build a pre-trained model that will be refined using unlabelled local energy datasets of the clients, thus offering a compromise between cloud-based and federated NILM algorithms. Moreover, it is widely acknowledged that standard FL schemes [46] are highly sensitive to data injection attacks. A study in this respect was conducted in [73] for the case of load disaggregation. The study revealed significant evidence of the vulnerability of federated load disaggregation where the operational states of appliances can easily be deduced and thus the daily activity. The adoption of blockchain technology is an up-and-coming solution in this regard. Yet, little to no attention was given to this aspect in related work.

Despite these limitations, federated NILM can be a prominent solution that combines the merits of edge-based model training and collaborative training through model exchange. However, we highlight that the nonavailability of suitable toolkits remains a major obstacle in this regard. More precisely, NILM toolkits fall short in keeping up with on-the-point technologies and frameworks [21], including FL. In addressing this gap, we suggest an extension of the most recent NILM toolkit Deep-NILMtk [21], with a federated trainer fully compatible with NILMtk and seq2point models implemented in the toolkit, that is, three baselines and the UNET-NILM model [74]. Our goal is to provide NILM scholars with a tool that would facilitate experimentation with different aspects and scenarios of FL. However, further efforts are also required (e.g., implementing different aggregation algorithms).

## 9. Conclusions

The current study provided a meta-analysis of load disaggregation intending to address three main research questions. The first one concerns the available taxonomies, where our analysis demonstrated that two main taxonomies exist for neural NILM: (i) based on features and (ii) based on the learning algorithm, considered a subclass of

the first one. As far as NILM concepts are considered, we identify that some concepts were more investigated in recent reviews, including feature extraction, learning algorithms, and evaluation setups. Preprocessing and postprocessing techniques, along with FL and computing platforms, received less attention. A main limitation of this first part is that all the findings are highly related to the set of selected systematic reviews. Consequently, we provided an overview of federated NILM approaches in the second part. Among FL aspects that were investigated in the scope of load disaggregation, the non-IIDness nature of energy data, the aggregation algorithms, and differential privacy were among the most frequent. As far as the aggregation performance is considered, federated load disaggregation provided acceptable results. Yet, they remain relatively low compared to values reported in quantitative analysis of recent NILM reviews. The non-IIDness of the data remains the main challenge in this regard.

To further investigate other aspects of FL, the current manuscript presented a case study considering different buildings of the REFIT data set. Even though limited, the results and the findings obtained from the simulation study demonstrate promising results for efficiently exploiting resources available on the edge while delivering acceptable disaggregation results on both seen and unseen buildings. Finally, considering both centralized and FL approaches, a summary of the main challenges yet to be addressed in NILM research was presented.

**Author Contributions:** Conceptualization, H.B., Y.H. and I.V.; Methodology, H.B., Y.H. and I.V.; Writing—original draft, H.B., Y.H. and I.V.; Writing—review & editing, F.B. and A.A.; Supervision, F.B. All authors have read and agreed to the published version of the manuscript.

**Funding:** This research received no external funding.

**Conflicts of Interest:** The authors declare no conflict of interest.

## Abbreviations

The following abbreviations are used in this manuscript:

| | |
|---|---|
| AI | Artificial Intelligence |
| DL | Deep Learning |
| ML | Machine Learning |
| DNN | Deep Neural Networks |
| FL | Federated Learning |
| MAE | Mean Average Error |
| NN | Neural Networks |
| NILM | Non-Intrusive Load Monitoring |

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
