# Peer review of "Neural Load Disaggregation: Meta-Analysis, Federated Learning and Beyond"

_energies, doi:10.3390/en16020991_

Round 1
Reviewer 1 Report
please see the attached

Reviewer 2 Report
The paper is of great interest, summarizes a great amount of information, which is then presented in a very structured way. Only a few comments:
1. From a literature review perspective, I would like to see some data categorization depending on the data resolution, i.e. typical smart meter data (15/30/60 min resolution) to sec-data or sub-sec data. The granularity is most of the times the key to unlock appliance detection in an accurate way.
2. With regards to training, it would be nice to distinguish between installation (home) specific appliance models and/or generic appliance models that are used for all houses.
3. An additional challenge, often neglected, is that even in the cases ground truth data exist that's most of the times for specific only appliances, e.g. fridge, washing machine etc. But nowadays, most of the energy consumption derives from appliances that are not attached to a plug (EV, heat pumps, AIrCos) which makes it even more complicated.
Round 2
Reviewer 1 Report
The authors have reviewed the manuscript intensively and the improvements can clearly be seen. Especially Figure 2 is now a valuable contribution. However, a few more things should be addressed:
1) Introducing Chapter 7 which has a total of 13 lines of text is not really meaningful. The information can be easily put elsewhere with a reference to the published code
2) Figure 3 does not really fit the style of the manuscript (I assume its drawn with the standard Visio settings) it should be in line with the rest of the graphics
3) Table 2 should be integrated into a vertical page layout. This can be easily done by reducing the space after the bullet points.
4) Personally I am not sure how valuable Chapter 8 is. The manuscript would be more comprehensive without it and the information could be easily placed in the code repository. This is a suggestion
5) I final spell and grammar checking should be done
